# Biorecovery of Agricultural Soil Impacted by Waste Motor Oil with *Phaseolus vulgaris* and *Xanthobacter autotrophicus*

**DOI:** 10.3390/plants11111419

**Published:** 2022-05-26

**Authors:** Blanca Celeste Saucedo Martínez, Liliana Márquez Benavides, Gustavo Santoyo, Juan Manuel Sánchez-Yáñez

**Affiliations:** Laboratorio de Microbiología Ambiental, Instituto de Investigaciones Químico-Biológicas Ed-B3, Universidad Michoacana de San Nicolás de Hidalgo, Morelia 58060, Mich, Mexico; 0617797j@umich.mx (B.C.S.M.); liliana.marquez@umich.mx (L.M.B.); gustavo.santoyo@umich.mx (G.S.)

**Keywords:** soil, hydrocarbons, cometabolism, phytotoxicity, legume, endophytic bacteria

## Abstract

Agricultural soil contamination by waste motor oil (WMO) is a worldwide environmental problem. The phytotoxicity of WMO hydrocarbons limits agricultural production; therefore, Mexican standard NOM-138-SEMARNAT/SSA1-2012 (NOM-138) establishes a maximum permissible limit of 4400 ppm for hydrocarbons in soil. The objectives of this study are to (a) biostimulate, (b) bioaugment, and (c) phytoremediate soil impacted by 60,000 ppm of WMO, to decrease it to a concentration lower than the maximum allowed by NOM-138. Soil contaminated with WMO was biostimulated, bioaugmented, and phytoremediated, and the response variables were WMO concentration, germination, phenology, and biomass of *Phaseolus vulgaris*. The experimental data were validated by Tukey HSD ANOVA. The maximum decrease in WMO was recorded in the soil biostimulated, bioaugmented, and phytoremediated by *P. vulgaris* from 60,000 ppm to 190 ppm, which was considerably lower than the maximum allowable limit of 4400 ppm of NOM-138 after five months. Biostimulation of WMO-impacted soil by detergent, mineral solution and bioaugmentation with *Xanthobacter autotrophicus* accelerated the reduction in WMO concentration, which allowed phytoremediation with *P. vulgaris* to oxidize aromatic hydrocarbons and recover WMO-impacted agricultural soil faster than other bioremediation strategies.

## 1. Introduction

Agricultural soils are affected by mixtures of aliphatic and aromatic hydrocarbons from waste motor oil (WMO) because of spills that limit plant growth and health [1,2,3]. The maximum recognized environmental standard in Mexico, NOM-138 SEMARNAT/SSA1-2012 (NOM-138) [4], establishes a maximum permissible limit of 4400 ppm for hydrocarbons in soil.

Globally, agricultural soils are polluted with high levels of WMO [5]. When agricultural soil is contaminated, plants can absorb hydrocarbons, inhibit plant growth, and decrease yield over long periods. These agricultural products pose a risk to human health [6]. An ecological alternative to solve this problem is biostimulation with detergent solutions such as Tween^®^ 80 (Sigma-Aldrich, Darmstadt, Germany) and Triton^®^ X-100 (Sigma-Aldrich, Darmstadt, Germany) to emulsify WMO, inducing heterotrophic soil microbiota by enrichment with mineral solution (MISO) containing essential elements to decrease WMO concentration [7]. 

Subsequent bioaugmentation occurs when the soil is inoculated with *Xanthobacter autotrophicus,* a facultative endophytic bacteria that uses WMO as a source of carbon and energy to remove WMO [8,9,10,11,12,13,14,15,16]. Biostimulation and bioaugmentation are strategies for reducing WMO and phytoremediation by *Phaseolus vulgaris* and *X. autotrophicus* [17] to ensure that the final WMO concentration is below the limit accepted by NOM-138 [4]. The objectives of this study were (a) biostimulation, (b) bioaugmentation, and (c) phytoremediation of soil impacted by 60,000 ppm of WMO to decrease it to a concentration lower than the maximum allowed by NOM-138 [4].

## 2. Results

### 2.1. Physicochemical Analysis of the Agricultural Soil

Table 1 shows the physicochemical analysis of the agricultural soil. 

### 2.2. Reduction in Waste Motor Oil after Biostimulation and Soil Bioaugmentation

Table 2 shows the decreasing concentration of WMO from 60,000 to 17,599 ppm in agricultural soil (70.6% WMO mineralization) when biostimulated with Tween^®^ 80, Triton^®^ X-100, and 50% of MISO followed by bioaugmentation with *X. autotrophicus*. Results of Tukey HSD test (*p* < 0.05) confirmed that these values were significantly lower than those for the biostimulation of WMO-polluted soil from 60,000 to 23,088 ppm (61.52% of WMO mineralization). These values were also significantly lower than those for the agricultural soil contaminated by WMO not subjected to biostimulation or bioaugmentation (negative control) with 39,215 ppm of WMO from the initial 60,000 ppm (34.65% of WMO mineralization). 

### 2.3. Germination of P. vulgaris with X. autotrophicus in Phytoremediation of Soil

Table 3 shows the 92% maximum germination percentage of *P. vulgaris* with *X. autotrophicus* after five days of phytoremediation of agricultural soil polluted by WMO. This percentage numerical value was significantly higher than the 83% germination of *P. vulgaris* not inoculated with *X. autotrophicus* compared to 100% germination of *P. vulgaris* in soil not polluted by WMO, fed with 100% MISO used as a relative control with 80% germination. 

### 2.4. Phenology and Biomass of P. vulgaris Enhanced with X. autotrophicus in Soil Phytoremediation at the Seedling Stage

Table 4 shows the phenology and biomass of *P. vulgaris* at the seedling stage during the phytoremediation of agricultural soil affected by WMO. *P. vulgaris* reached the greatest plant height (PHE) of 36.42 cm, with an aerial fresh weight (AFW) of 2.13 cm, and aerial dry weight (ADW) of 0.219 cm when planted in soil previously biostimulated and bioaugmented. The above numerical values were significantly higher than those of *P. vulgaris* cultured in soil not contaminated by WMO, fed with 100% MISO, and used as a relative control with 31.27 cm of PHE, 1.82 g of AFW, and 0.16 g of ADW and *P. vulgaris* grown in agricultural soil not contaminated by WMO irrigated only with water used as an absolute control with a root length (RL) of 7.16 cm, root fresh weight (RFW) of 0.1 g, and root dry weight (RDW) of 0.016 g. The above numerical values were significantly higher than those of *P. vulgaris* grown in agricultural soil contaminated by WMO and uninoculated with *X. autotrophicus*, previously biostimulated and bioaugmented with 5.85 cm of RL, 0.04 g RFW, and 0.011 g RDW. 

### 2.5. Phenology and Biomass of P. vulgaris Enhanced with X. autotrophicus in Soil Phytoremediation at the Physiological Maturity Stage 

Table 5 shows the phenology and biomass of *P. vulgaris* at the physiological maturity stage during phytoremediation of agricultural soil polluted by WMO, as can be seen in Figure 1. *P. vulgaris* showed PHE of 108.57 cm, RL of 20.28 cm, AFW of 8.37 g, RFW of 3.61 g, ADW of 1.12 g, and RDW of 0.21 g. The above numerical values were significantly higher than those of *P. vulgaris* fed with 100% MISO growing in soil uncontaminated by WMO (relative control) with 70.56 of PHA, 18.25 cm of RL, 3.78 g of AFW, 1.24 g of RFW, 0.72 g of ADW, 0.08 g RDW. Similarly, *P. vulgaris* enhanced with *X. autotrophicus* showed 93.38 cm of PH, 17.33 cm of RL, 4.58 g AFW, 1.51 g of RFW, 0.63 g ADW, and 0.1 g of RDW. 

### 2.6. Final WMO Concentration in Agricultural Soil after Biostimulation, Bioaugmentation, and Phytoremediation

Table 6 shows the final WMO concentration in agricultural soil after three months of biostimulation and bioaugmentation, after two months of phytoremediation. The maximum decrease in WMO concentration was detected in the soil biostimulated, bioaugmented, and phytoremediated with *P. vulgaris* where WMO decreased from 60,000 to 190 ppm, a value that was significantly lower than that for soil contaminated by WMO. Biostimulation, bioaugmentation, and phytoremediation with *P. vulgaris* enhanced with *X. autotrophicus* improved the WMO reduction from 60,000 to 680 ppm; the agricultural soil affected by WMO only biostimulated and phytoremediated with *P. vulgaris* showed a final concentration of 3040 ppm from an initial of 60,000 ppm. All WMO numerical concentration values were lower than the accepted maximum limit of 4400 ppm of NOM 138, and at the same time were significantly lower than that for the 35,210 ppm WMO of the negative control soil.

### 2.7. Residual Hydrocarbons in WMO after Biostimulation, Bioaugmentation, and Phytoremediation

The chromatogram (Figure 2) shows that after three months of biostimulation and bioaugmentation of soil impacted by WMO, the simplest aliphatic hydrocarbons, linear, branched short- and long-chain hydrocarbons of 12 and 14 carbons, linear hydrocarbons consisting of up to 35 carbons, and aromatics such as (1-methyl-4-(1-methylpropyl)-benzene, 2-nitro-tertiary-butanol, and 2-isopropyl-5-methyl-1 heptanol were eliminated. Aliphatics such as nonadecane, eneicosane, eicosane, and heptacosane were also eliminated, while aromatics, naphthalene, and 1-cyclohex-1-1-enyl-1-phenyl-ethanol were oxidized. The remaining hydrocarbons were some of the more complex 14-carbon and 36-carbon aromatics as well as some 17-carbon aliphatics.

## 3. Discussion

The physicochemical properties of the soil (Table 1) showed that the soil required biostimulation with 50% MISO to ensure that the native soil microorganisms received the maximum concentration of basic minerals to induce WMO mineralization [18].

The decrease in WMO after biostimulation and bioaugmentation (Table 2) indicates that biostimulation and bioaugmentation accelerated mineralization of WMO, as has been reported in another research [19]. First, biostimulation with a mix of detergents emulsified the WMO and 50% MISO to enable degradation of WMO by the native heterotrophic aerobic microorganisms. WMO removal was then improved by bioaugmentation with *X. autotrophicus* to decrease the concentration of WMO and facilitate phytoremediation of soil with *P. vulgaris* and *X. autotrophicus* to reduce the concentration of WMO to a value below the maximum accepted value of NOM-138 [4].

The germination percentage (Table 3) results suggested that *P. vulgaris* improved with *X. autotrophicus* increased tolerance to WMO and, therefore, provided maximum germination. While *P. vulgaris* not inoculated with *X. autotrophicus* showed symptoms of phytotoxicity to WMO and, therefore, showed a reduction in germination percentage, similar to observations reported in previous research where excess hydrocarbons inhibited seed germination for soil phytoremediation [20,21,22].

The growth of *P. vulgaris* at seedling level (Table 4) showed its sensitivity to WMO phytotoxicity; this response of *P. vulgaris* is used as an indicator of the degree of recovery or contamination of the soil to the concentration of WMO [15,23]. Thus, if the WMO concentrations in soil decrease, *P. vulgaris* starts growth with a pattern analogous to that observed when the soil is not contaminated with WMO in part because bioaugmentation with *X. autotrophicus* cimulation, bioaugmentation, and phytoremediaauses oxidation of aromatic hydrocarbons and, subsequently, reduction of the concentration of WMO as reported in the literature [17,19,24].

The biomass values of *P. vulgaris* at physiological maturity (Table 5) indicate healthy growth only in agricultural soils without hydrocarbons (Figure 1). This result supports the fact that phytoremediation of any agricultural soil could be applied to soils contaminated with relatively high concentration of hydrocarbons such as those of WMO, if its biorecovery begins with biostimulation followed by bioaugmentation, where *X. autotrophicus* can decrease WMO concentration and finally when *P. vulgaris* is enhanced with *X. autotrophicus,* a facultative endophyte able to invade roots to transform organic compounds of photosynthesis into phytohormones [15,25]. *X. autotrophicus* induces a root system in *P. vulgaris* with a greater capacity for the phytodegradation of WMO and simultaneously optimizes the absorption of minerals [26,27]. The above facilitates the biorecovery of agricultural soil and allows it to be safely implemented in agricultural cultivation, while ensuring that the produce obtained is safe for human and animal consumption due to a decrease in WMO equivalent to any soil free of WMO contamination [21,28].

The response of *P. vulgaris* with and without *X. autotrophicus* shows that it is possible to biorecover agricultural soil contaminated by a relatively high concentration of WMO (Table 6) [12,22,26,27]. However, it was evident that biostimulation of this soil was affected by WMO without bioaugmentation [29], followed by phytoremediation with only *P. vulgaris*. Although it reduced the concentration of WMO to a value of 3040 ppm, which was lower than the value limit recognized by NOM-138 [4], the treatment does not fully render the use of the agricultural soil for plant production because of the risk of absorption of the WMO by agricultural crops [1,2,30]. In clear contrast to the biostimulation of the soil impacted by WMO, followed by bioaugmentation with *X. autotrophicus*, which has the ability to degrade aromatics from the WMO [14,16] to continue with the phytoremediation of the WMO with *P. vulgaris,* which was housed by *X. autotrophicus* [20,25], a decrease of 190 ppm in the WMO was achieved [12,31], a concentration equivalent to that observed in soil devoid of contamination by hydrocarbon mixtures that ensures agricultural production without risk to the human or animal consumer [2,18,30]. The decrease in the concentration of WMO in the agricultural soil when biostimulated, followed by bioaugmentation with *X. autotrophicus* and then inoculation of *P. vulgaris* in phytoremediation, resulted in a final concentration of 680 ppm of WMO [32]. These concentrations are still in the concentration range detected in soil not contaminated with hydrocarbon mixtures; thus, agricultural soil is safe for agricultural production without the risk arising from consumption of produce [1]. It also indicates that the presence of an excess of *X. autotrophicus* in the soil, as in *P. vulgaris*, does not improve its ability to remove WMO [12,33]. In the agricultural soil used as a negative control, the WMO decrease was due to natural attenuation caused by soil microorganisms able to mineralize WMO according to environmental conditions depending on time [9].

The remaining hydrocarbons detected in the chromatogram (Figure 2) indicate that soil biostimulation and bioaugmentation accelerated the phytoremediation process of the impacted soil to remove most of the remaining WMO hydrocarbons by *P. vulgaris*, indicating that the use of *P. vulgaris* in association with native microorganisms and the endophyte *X. autotrophicus* removed not only aliphatic hydrocarbons but also many aromatics in the soil bioaugmentation, which maximized the removal of WMO and shortened the soil recovery time.

## 4. Materials and Methods

### 4.1. Agricultural Soil Sample Collection and Preparation

Soil was collected from a mild weather site in the municipality of Salvador Escalante, Mich, Mexico. Physicochemical analysis of the soil was performed according to the NOM-021-SEMARNAT-2000 standard [34]. The soil was solarized at 70 °C/48 h to minimize the problem of pests and diseases and sieved with No. 20 mesh.

### 4.2. Biostimulation and Bioaugmentation of Soil Impacted by WMO with X. autotrophicus

A sextuplicate, randomized block experiment was conducted in Leonard jars with 1 kg of soil. The soil was contaminated with 60,000 ppm of the WMO trademark Mobil^®^ ExxonMobil, Irving, TX, USA from a mechanical workshop. The initial WMO concentration was corroborated by Soxhlet analysis, which is widely used in the literature [27]. The first step of biostimulation began with a 0.5% solution of detergents mixed with Tween^®^ 80 and Triton^®^ X-100, both Sigma Aldrich^®^ (Darmstadt, Germany) and laboratory grade. The solution of these detergents at low concentrations, such as 0.5%, does not inhibit soil microbiota and promotes WMO hydrocarbon emulsification [35,36]. Simultaneously, the soil was biostimulated with 50% MISO, where the nutrient amendment was made to achieve an N:P ratio of 10:1, which is the recommended ratio range to promote and enhance hydrocarbon mineralization [37], with the following composition (g/L): NH_4_NO_3_ 10, K_2_HPO_4_ 2.5, KH_2_PO_4_ 2, MgSO_4_ 1, NaCl 0.1, CaCl_2_ 0.1, FeSO_4_ traces, element solution 10, and pH 6.5–6.8. The second step for soil polluted by WMO was bioaugmentation with *X. autotrophicus* that was activated on agar without nitrate and sucrose with the following composition (g/L): K_2_HPO_4_ 1, MgSO_4_-H_2_O 0.5, KCl 0.5, FeSO_4_.7H_2_O, pH 7.2 [25], and incubated at 30 °C for 48 h. *Xanthobacter autotrophicus* was obtained from Collection of Department of Chemistry and Chemical Biology, Harvard University, 12 Oxford Street, Cambridge, Massachusetts 02138, Cambridge, MA, USA, which had previously been isolated from polluted soil with hydrocarbons. The microbial cells of *X. autotrophicus* were suspended in detergent saline solution (NaCl 0.85% and detergent 0.01%) at a McFarland standard level of 0.5 containing approximately 1.5 × 10^8^ colony-forming units (CFU)/mL of *X. autotrophicus.* Then, the soil polluted by WMO was inoculated with 5 mL of *X. autotrophicus*/kg of soil, suspended in detergent saline solution for bioaugmentation according to the experimental design for three months with: absolute control of uncontaminated soil with only irrigated with water, a negative control of soil impacted by 60,000 ppm WMO, Treatment 1 (T1) soil + WMO, biostimulated with 50% of MISO, Tween^®^ 80, and 0.5% Triton^®^ X-100; Treatment 2 (T2) soil + WMO, biostimulated with 50% MISO, Tween^®^ 80, and 0.5% Triton^®^ X-100 and bioaugmented with *X. autotrophicus* (Figure 2). Measurement of the decrease in the WMO hydrocarbon concentration in the soil after biostimulation and bioaugmentation was performed as a total petroleum hydrocarbon decrease analysis using the Soxhlet extraction method. Soxhlet extraction is a classical method for the extraction of thermally stable and recalcitrant organic compounds from complex matrices [38,39,40].

### 4.3. Phytoremediation of Soil Impacted by WMO with P. vulgaris Enhanced with X. autotrophicus

Phytoremediation with *P. vulgaris* was enhanced with *X. autotrophicus* to obtain a maximum decrease of WMO still available after biostimulation and bioaugmentation in a sextuplicate in a randomized block experiment. *X. autotrophicus* was grown on agar without nitrate or sucrose [24]. Further, six uninoculated seeds of *P. vulgaris* were sown in each Leonard’s jar containing 1 kg of soil with WMO that was biostimulated with 50% MISO during phytoremediation (Figure 3), and 36 seeds of *P. vulgaris* were inoculated with 5.0 mL of *X. autotrophicus.* Its concentration was adjusted to the McFarland standard tube CFU 1 equivalent to 3 × 10^8^ colony-forming units/mL. The experimental design for two months was as follows: *P. vulgaris* in absolute control of uncontaminated soil only irrigated with water; *P. vulgaris* in relative control soil fed with 100% MISO; Treatment 1 (T1) *P. vulgaris* in soil + WMO, previously biostimulated with 50% MISO, Tween^®^ 80, and 0.5% Triton^®^ X-100; Treatment 2 (T2) *P. vulgaris* in soil + WMO, previously biostimulated with 50% MISO, Tween^®^ 80, and 0.5% Triton^®^ X-100 and bioaugmented with *X. autotrophicus,* and (T3) *P. vulgaris* inoculated with *X. autotrophicus* + WMO, previously biostimulated with 50% MISO, Tween^®^ 80, and 0.5% Triton^®^ X-100 and bioaugmented with *X. autotrophicus.* Germination percentage was measured at six days, and phenology and biomass were measured at the seedling and physiological maturity stages as indicators. After two months of phytoremediation, the final concentration of WMO was determined by total petroleum hydrocarbon decrease analysis using the Soxhlet extraction method [27,38,39,40]. Finally, soil samples with WMO were analyzed using an Agilent Technologies 7890A series gas chromatograph coupled to 5975C inert mass spectrometer (Conquer Scientific, Poway, CA, USA); 1.0 µL of the sample was injected in splitless mode, helium (99.995% purity) was used as the carrier gas in a Zebron-5MS capillary column with 30.0 m length, 0.25 mm internal diameter, and 0.25 mm film thickness. The injector temperature was 250 °C, and the initial oven temperature was 50 °C. The temperature detector was 280 °C with an equilibrium time of 3 min and a maximum temperature of 320 °C [41].

### 4.4. Statistical Analysis

The statistical estimation was carried out using Statgraphics Centurion XVI.II software. ANOVA was done with six replications for each combination of the nominal variables. In the case of statistically significant data (*p* < 0.05) a post hoc Tukey honest significant difference (HSD) was used [42,43].

## 5. Conclusions

Bioremediation of agricultural soils impacted by WMO, through biostimulation with MISO, detergent solution, and bioaugmentation with *X. autotrophicus*, accelerated the reduction of WMO concentration, which allowed phytoremediation with *P. vulgaris* enabling oxidation of aromatic hydrocarbons. This makes it an effective soil remediation strategy that ensures the reduction of WMO to a concentration similar to that detected in non-contaminated soils, allowing sustainable and safe agricultural production for humans and animal use in addition to showing rapid results compared to other bioremediation strategies.

## Figures and Tables

**Figure 1 plants-11-01419-f001:**
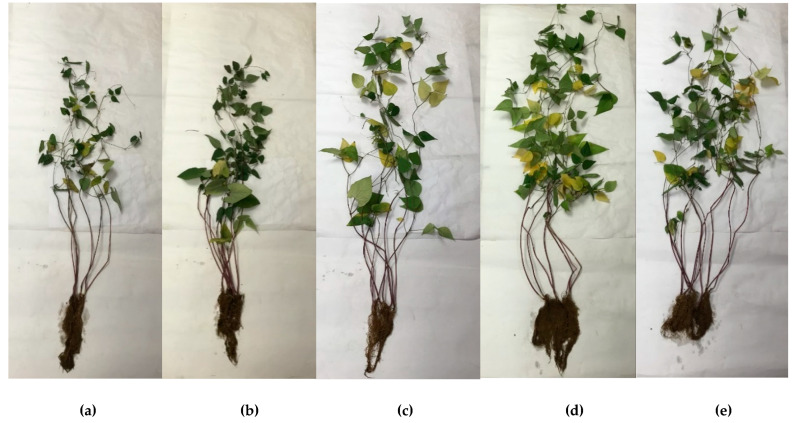
Growth of *Phaseolus vulgaris* enhanced with *Xanthobacter autotrophicus* at physiological maturity during phytoremediation of agricultural soil impacted by waste motor oil. (**a**) Absolute control *P. vulgaris* irrigated with water in agricultural soil devoid of waste motor oil (WMO) contamination. (**b**) Relative control *P. vulgaris* in agricultural soil devoid of WMO contamination, fed by 100% mineral solution (MISO). (**c**) T1 *P. vulgaris* in agricultural soil impacted by WMO biostimulated with Tween^®^ 80/ Triton^®^ X-100 and 50% MISO. (**d**) T2 *P. vulgaris* in agricultural soil impacted by WMO biostimulated with Tween^®^ 80/ Triton^®^ X-100 and 50% MISO and bioaugmented with *X. autotrophicus.* (**e**) T3 *P. vulgaris* enhanced with *X. autotrophicus* in agricultural soil impacted by WMO biostimulated with Tween^®^ 80/ Triton^®^ X-100, 50% MISO and bioaugmented with *X. autotrophicus*.

**Figure 2 plants-11-01419-f002:**
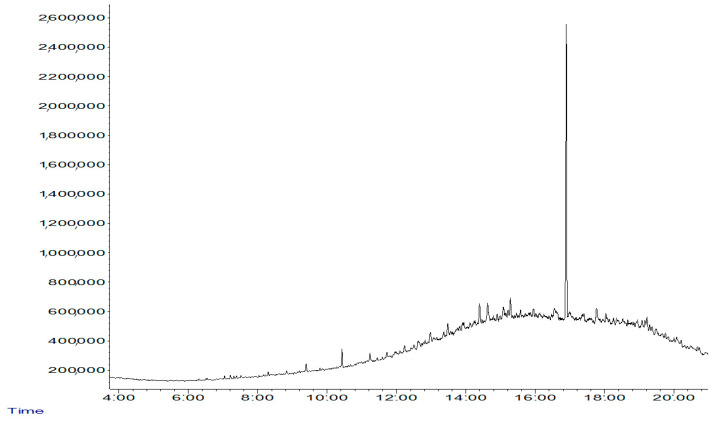
Chromatogram of 14 C, 36 C aromatic and 17 C aliphatic hydrocarbons in WMO after biostimulation, bioaugmentation, and phytoremediation.

**Figure 3 plants-11-01419-f003:**
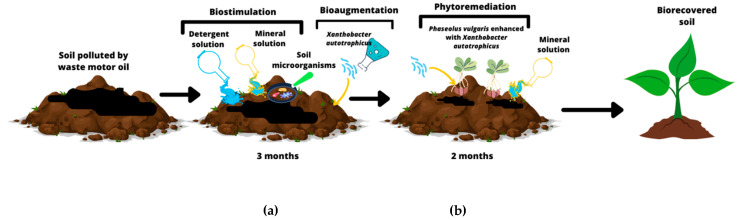
Bioremediation of soil impacted by waste motor oil, (**a**) biostimulation: induces mineralization of WMO by soil microorganisms/ bioaugmentation: increase WMO mineralization, (**b**) phytoremediation: end of the elimination of remaining WMO.

**Table 1 plants-11-01419-t001:** Physicochemical properties of agricultural soil not contaminated by waste motor oil.

Parameter	Value	Interpretation
pH	5.67	Moderately acidic (5.1–6.5)
Organic matter (%)	10.44	Very high (>6.0)
Texture (%)	31.8 (clay), 26.92 (sand), 42.0 (silt)	Clayey silt
Total nitrogen (ppm)	3200	Very high (>0.25)
Phosphorus (ppm)	219.34	Very high
Sodium (Na^+^) (ppm)	153.38	High
Potassium (K^+^) (ppm)	168.61	High
Microelements (ppm):		
Iron (Fe^2+^)	13.91	Appropriate (>4.5)
Zinc (Zn^2+^)	0.37	Deficient (<0.5)
Copper (Cu^2+^)	0.54	Appropriate (>0.2)
Manganese (Mn^2+^)	4.62	Appropriate ^1^ (>1)

^1^ NOM-021-SEMARNAT-2000.

**Table 2 plants-11-01419-t002:** Concentration of waste motor oil in agricultural soil after biostimulation and bioaugmentation with *Xanthobacter autotrophicus*.

Soil	WMO Concentration after 3 Months (ppm)	Mineralization Percentage (%)
Negative control soil impacted by WMO *	39,215 ± 0.06 ^c,^**	34.65 ± 0.01 ^c^
T1: Soil impacted by WMO biostimulated with Tween^®^ 80 + Triton^®^ X-100 and 50% MISO	23,088 ± 0.06 ^b^	61.62 ± 0.03 ^b^
T2: Soil impacted by WMO biostimulated with Tween^®^ 80/Triton^®^ X-100, 50% MISO and bioaugmented with *X. autotrophicus*	17,599 ± 0.12 ^a^	70.67 ± 0.24 ^a^

* WMO: waste motor oil; MISO: mineral solution. ANOVA Tukey *p* < 0.05 ** Different letters indicate significant difference, *n* = 6.

**Table 3 plants-11-01419-t003:** Germination percentage and days to emergency of *Phaseolus vulgaris* enhanced with *Xanthobacter autotrophicus* during phytoremediation of agricultural soil impacted by waste motor oil.

*Phaseolus vulgaris*	Emergency Days	Germination Percentage (%)
Absolute control *P. vulgaris* without WMO * soil irrigated only water	6 ± 0.13 ^b^^,^**	75 ± 0.15 ^c^
Relative control *P. vulgaris* without WMO fed 100% MISO	6 ± 0.02 ^b^	80 ± 0.10 ^b^
T1: *P. vulgaris* in soil impacted by WMO + biostimulated with Tween^®^ 80/Triton^®^ X-100 and 50% MISO	5 ± 0.03 ^a^	75 ± 0.01 ^c^
T2: *P. vulgaris* in soil impacted by WMO biostimulated with Tween^®^ 80/Triton^®^ X-100, 50% MISO and bioaugmented with *X. autotrophicus*	5 ± 0.08 ^a^	83 ± 0.03 ^b^
T3: *P. vulgaris* enhanced with *X. autotrophicus* in soil impacted by WMO biostimulated with Tween^®^ 80/Triton^®^ X-100, 50% MISO and bioaugmented with *X. autotrophicus*	5 ± 0.16 ^a^	92 ± 0.13 ^a^

* WMO: waste motor oil; MISO: mineral solution. ANOVA Tukey *p* < 0.05 ** Different letters indicate significant difference, *n* = 6.

**Table 4 plants-11-01419-t004:** Phenology and biomass of *Phaseolus vulgaris* enhanced with *Xanthobacter autotrophicus* at seedling level during phytoremediation of soil impacted by remainder waste motor oil.

*Phaseolus vulgaris*	Plant Height (cm)	Root Length (cm)	Aerial Fresh Weight (g)	Root Fresh Weight (g)	Aerial Dry Weight (g)	Root Dry Weight (g)
Absolute control *P. vulgaris* without WMO * soil irrigated only water	27.25 ± 0.02 ^c,^**	7.16 ± 0.06 ^a^	1.46 ± 0.18 ^d^	0.1 ± 0.14 ^a^	0.161 ± 0.14 ^c^	0.016 ± 0.10 ^a^
Relative control *P. vulgaris* without WMO fed 100% MISO	31.27 ± 0.10 ^b^	6.5 ± 0.23 ^b^	1.82 ± 0.27 ^b^	0.09 ± 0.05 ^b^	0.16 ± 0.05 ^c^	0.012 ± 0.18 ^c^
T1: *P. vulgaris* in soil impacted by WMO + biostimulated with Tween^®^ 80/Triton^®^ X-100 and 50% MISO	30 ± 0.15 ^b^	4.14 ± 0.27 ^d^	1.39 ± 0.18 ^d^	0.04 ± 0.30 ^d^	0.169 ± 0.30 ^c^	0.009 ± 0.15 ^e^
T2: *P. vulgaris* in soil impacted by WMO biostimulated with Tween^®^ 80/Triton^®^ X-100, 50% MISO and bioaugmented with *X. autotrophicus*	36.42 ± 0.16 ^a^	5.85 ± 0.05 ^c^	2.13 ± 0.15 ^a^	0.04 ± 0.18 ^d^	0.219 ± 0.23 ^a^	0.011 ± 0.06 ^d^
T3: *P. vulgaris* enhanced with X. autotrophicus in soil impacted by WMO biostimulated with Tween^®^ 80/Triton^®^ X-100, 50% MISO and bioaugmented with *X. autotrophicus*	27.25 ± 0.18 ^c^	6.58 ± 0.20 ^b^	1.62 ± 0.10 ^c^	0.06 ± 0.22 ^c^	0.184 ± 0.10 ^b^	0.014 ± 0.16 ^b^

* WMO: waste motor oil; MISO: mineral solution. ANOVA Tukey *p* < 0.05 ** Different letters indicate significant difference, *n* = 6.

**Table 5 plants-11-01419-t005:** Phenology and biomass of *Phaseolus vulgaris* enhanced with *Xanthobacter autotrophicus* at physiological maturity level during phytoremediation of soil impacted by waste motor oil.

*Phaseolus vulgaris*	Plant Height (cm)	Root Length (cm)	Aerial Fresh Weight (g)	Root Fresh Weight (g)	Aerial Dry Weight (g)	Root Dry Weight (g)
Absolute control *P. vulgaris* without WMO * soil irrigated only with water	76.42 ± 0.07 ^d,^**	20.42 ± 0.02 ^a^	2.4 ± 0.15 ^e^	1.38 ± 0.23 ^d^	0.71 ± 0.03 ^c^	0.14 ± 0.15 ^b^
Relative control *P. vulgaris* without WMO fed 100% MISO	70.56 ± 0.12 ^e^	18.25 ± 0.18 ^b^	3.78 ± 0.23 ^d^	1.24 ± 0.11 ^e^	0.72 ± 0.13 ^c^	0.08 ± 0.22 ^d^
T1: *P. vulgaris* in soil impacted by WMO + biostimulated with Tween^®^ 80/Triton^®^ X-100 and 50% MISO	84.85 ± 0.1 ^c^	17.28 ± 0.12 ^c^	4.18 ± 0.15 ^c^	1.79 ± 0.08 ^d^	0.91 ± 0.24 ^b^	0.12 ± 0.14 ^c^
T2: *P. vulgaris* in soil impacted by WMO biostimulated with Tween^®^ 80/Triton^®^ X-100, 50% MISO and bioaugmented with *X. autotrophicus*	108.57 ± 0.13 ^a^	20.28 ± 0.35 ^a^	8.37 ± 0.06 ^a^	3.61 ± 0.15 ^a^	1.12 ± 0.31 ^a^	0.21 ± 0.04 ^a^
T3: *P. vulgaris* enhanced with *X. autotrophicus* in soil impacted by WMO biostimulated with Tween^®^ 80/Triton^®^ X-100, 50% MISO and bioaugmented with *X. autotrophicus*	93.38 ± 0.12 ^b^	17.33 ± 0.40 ^c^	4.58 ± 0.01 ^b^	1.51 ± 0.02 ^c^	0.63 ± 0.10 ^d^	0.1 ± 0.14 ^c^

* WMO: waste motor oil; MISO: mineral solution. ANOVA Tukey *p* < 0.05 ** Different letters indicate significant difference *n* = 6.

**Table 6 plants-11-01419-t006:** Final concentration of waste motor oil in agricultural soil after three months of biostimulation, bioaugmentation, and two months of phytoremediation.

Soil	WMO Final Concentration (ppm)
Negative control soil impacted by WMO *	35,210 ± 0.12 ^d^
T1: Soil impacted by WMO biostimulated with Tween^®^ 80/Triton^®^ X-100, 50% MISO and phytoremediated with *P. vulgaris*	3040 ± 0.01 ^c^
T2: Soil impacted by WMO biostimulated with Tween^®^ 80/Triton^®^ X-100, 50% MISO, bioaugmented with *X. autotrophicus* and phytoremediated with *P. vulgaris*	190 ± 0.03 ^a,^**
T3: Soil impacted by WMO biostimulated with Tween^®^ 80/Triton^®^ X-100, 50% MISO, bioaugmented with *X. autotrophicus,* and phytoremediated with *P. vulgaris* inoculated with *X. autotrophicus*	680 ± 0.09 ^b^

* WMO: waste motor oil; MISO: mineral solution. ANOVA Tukey *p* < 0.05 ** Different letters indicate significant difference, *n* = 6.

## Data Availability

Not applicable.

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
