# Peer review of "Biorecovery of Agricultural Soil Impacted by Waste Motor Oil with *Phaseolus vulgaris* and *Xanthobacter autotrophicus"

_plants, 2022, doi:10.3390/plants11111419_

Round 1

Reviewer 1 Report

Bioremediation  of soils contaminated with used engine oil is one of the methods already used on a large scale. However, several questions remain open regarding the operational conditions and biological strategies that should be adopted to optimize removal efficiency. The manuscript "Bio-recovery of agricultural soils affected by waste engine oil from Phaseolus vulgaris and Xanthobacter autotrophicus" seems to be interesting for the public opinion of "Plants", because the adopted research strategy consisting in combining biostimulation processes with bioaugmentation and phytoremediation is the appropriate approach which allows to achieve the assumed reduction targets the content of waste engine oil in the soil.However, the reviewer has serious reservations as to the quality of the research and its interpretation, and the article is currently not suitable for publication in the journal "Plants".·     

  • The authors should first of all revise the methodology of determining the content of waste engine oil (WMO), allowing to determine its composition (the preferred chromatographic method), which will allow for the assessment of the effectiveness of individual bioremediation processes and drawing appropriate conclusions.·       
  • It is advisable to present the course of bioremediation processes in a graphic form, as it may familiarize the reader with the intricate way of its presentation and remove errors in defining the time frames of individual stag es of bioremediation in the text and tables.·       
  • The bioaugmentation process with X. autotrophicus is presented in a very laconic way. 

Below are some detailed comments that may be helpful to the authors. 

Line 36. Two types of detergent nomenclature were used in the text: Triton X-100 and Tritón X-100. Please standardize the nomenclature throughout the manuscript

Bacterial names: Xanthobacter autotrophicus and plants: Phaseolus vulgaris should be in italics. Check such mistakes along with the manuscript.

Please provide information from where the bacterium Xanthobacter autotrophic was isolated from. How was it identified.

Line 51. The nitrogen content in the soil is given in %, and the phosphorus in ppm.Please standardize the units.

The ratio of nitrogen to phosphorus is important in the degradation of petroleum pollutants. What ratio of nitrogen to phosphorus was used in the soil during bioaugmentation. Why did you use this nitrogen to phosphorus ratio?

Lines 123-124. The description shows that the process of biostimulation and bioagmentation lasted 3 months and phytoremediation - 2 months. The table shows that the process of biostimulation, bioagmentation and phytoremediation lasted 3 months. How long did the entire remediation process take? The description is not clear.

Line 139. The % sign should be after the numeric value with no spaces. Please correct any mistakes in the manuscript

Line 203. Please change análisis to analisis

Line 505. Change N° to No.

The tables do not contain the values of the marking errors. Please enter error values in the form ± in tables or in the description below the tables. The explanation provided is practically unverifiable, as the methodology for statistics refers to a literature item of over 800 pages.

Line 217. Is the 1 x 108 cells/mL is correct? Shouldn't there be 1 x 108 CFU/mL

Line 224 and line 235. Please describe the methodology for marking WMO. Sentence: The response variable of soil biostimulated and bioaugmented was the final concentration of WMO by Soxhlet is incorrect. Where reference is made to the methodology described in the article, the author of the article should be given. In addition, the Soxhlet method is used to extract analytes, not to determine them. If you do not intend to accurately describe the WMO determination method, it is worth referring to the standard according to which they were labeled. Reference 35 is a review article that describes many of the methods used in the determination of petroleum hydrocarbons, therefore the reader cannot verify your method. In a scientific article, the description of the methodology should be clear and lucid, so that the reader knows exactly what research has been done.

Line 231. The acronym CFU stands for colony forming unit, not colony farming unit

Line 248- 250. :In the conclusion section, the authors suggest that aromatic hydrocarbons are degraded as a result of the applied remediation treatments. These conclusions are not supported by research. The manuscript did not identify individual groups of hydrocarbons, and only focused on the concentration of waste motor oil (WMO). In addition, the exact composition of the WMO used to contaminate the soil was not given. The manuscript only states that it is a mixture of hydrocarbons

Author Response

Please also see the attachment 

Point 1. The authors should first of all revise the methodology of determining the content of waste engine oil (WMO), allowing to determine its composition (the preferred chromatographic method), which will allow for the assessment of the effectiveness of individual bioremediation processes and drawing appropriate conclusions.· 

Response 1. The details of the requested methodology were detailed in lines 233-234 page 8, line 259 page 8, line 261-263 page 9. Lines 284-294 page 9.

Point 2. It is advisable to present the course of bioremediation processes in a graphic form, as it may familiarize the reader with the intricate way of its presentation and remove errors in defining the time frames of individual stag es of bioremediation in the text and tables.

Response 2. A graphic on the bioremediation process was added on line 296 page 9.

Point 3. The bioaugmentation process with X. autotrophicus is presented in a very laconic way. Below are some detailed comments that may be helpful to the authors. 

Response 3. Added details of the bioaugmentation process on lines 245-254, page 8.

Point 4. Line 36. Two types of detergent nomenclature were used in the text: Triton X-100 and Tritón X-100. Please standardize the nomenclature throughout the manuscript

Response 4. The nomenclature of detergents has already been standardized throughout the text

Point 5. Bacterial names: Xanthobacter autotrophicus and plants: Phaseolus vulgaris should be in italics. Check such mistakes along with the manuscript.

Response 5. All bacterial names have already been written in italics.

Point 6. Please provide information from where the bacterium Xanthobacter autotrophic was isolated from. How was it identified.

Response 6. Information on the origin of the bacteria has already been added to lines 245- 248, page 8.

Point 7. Line 51. The nitrogen content in the soil is given in %, and the phosphorus in ppm.Please standardize the units.

Response 7. The ppm units have already been standardized in Table 1, line 52, page 2.

Point 8. The ratio of nitrogen to phosphorus is important in the degradation of petroleum pollutants. What ratio of nitrogen to phosphorus was used in the soil during bioaugmentation. Why did you use this nitrogen to phosphorus ratio?

Response 8. Information on the N:P ratio has already been added on lines 237-240, page 8.

Point 9. Lines 123-124. The description shows that the process of biostimulation and bioagmentation lasted 3 months and phytoremediation - 2 months. The table shows that the process of biostimulation, bioagmentation and phytoremediation lasted 3 months. How long did the entire remediation process take? The description is not clear.

Response 9. The process lasted 3 months of biostimulation and bioaugmentation, and finally 2 months of phytoremediation. This information is already standardized in Table 2, line 63 page 2, Table 6, line 134 page 5, line 254 page 8, line 284 and 296 page 9.

Point 10. Line 139. The % sign should be after the numeric value with no spaces. Please correct any mistakes in the manuscript

Response 10. The % sign has already been separated from the numbers throughout the text.

Point 11. Line 203. Please change análisis to análisis

Response 11. Ensured the correct spelling of the word "analysis" on lines 49, 50, page 2, line 261 page 9, line 286 page 9, and 304 page 10.

Point 12. Line 505. Change N° to No.

Response 12. N° has already been changed to No. on line 229 page 8.

Point 13. The tables do not contain the values of the marking errors. Please enter error values in the form ± in tables or in the description below the tables. The explanation provided is practically unverifiable, as the methodology for statistics refers to a literature item of over 800 pages.

Response 13. Values of the marking errors were added to the tables. Also the reference of the statistical part was changed on line 306 page 10 and line 447 page 12.

Point 14. Line 217. Is the 1 x 108 cells/mL is correct? Shouldn't there be 1 x 108 CFU/mL

Response 14. CFU units were correctly written on lines 252 page 8, lines 274-275 page 9.

Point 15. Line 224 and line 235. Please describe the methodology for marking WMO. Sentence: The response variable of soil biostimulated and bioaugmented was the final concentration of WMO by Soxhlet is incorrect. Where reference is made to the methodology described in the article, the author of the article should be given. In addition, the Soxhlet method is used to extract analytes, not to determine them. If you do not intend to accurately describe the WMO determination method, it is worth referring to the standard according to which they were labeled. Reference 35 is a review article that describes many of the methods used in the determination of petroleum hydrocarbons, therefore the reader cannot verify your method. In a scientific article, the description of the methodology should be clear and lucid, so that the reader knows exactly what research has been done.

Response 15. The initial WMO concentration was corroborated by Soxhlet analysis widely used in the literature in lines 233-234 page 8. Information on the use of the Soxhlet method for the determination of the WMO concentration widely accepted by Unites States Environmental Protection Agency (USEPA), and used in the literature was added on lines 259-263 pages 8 and 9, lines 284-287 page 9. In addition, a chromatographic analysis was added at the end to determine the final hydrocarbons in lines 287-294 page 9. The reference 35 was changed for others in lines 263 and 287, page 9. Lines 407-409 page 11 and lines 438-448, page 12.

Point 16. Line 231. The acronym CFU stands for colony forming unit, not colony farming unit

Response 16. CFU units were correctly written on lines 252 page 8, lines 274-275 page 9.

Point 17. Line 248- 250: In the conclusion section, the authors suggest that aromatic hydrocarbons are degraded as a result of the applied remediation treatments. These conclusions are not supported by research. The manuscript did not identify individual groups of hydrocarbons, and only focused on the concentration of waste motor oil (WMO). In addition, the exact composition of the WMO used to contaminate the soil was not given. The manuscript only states that it is a mixture of hydrocarbons

Response 17.  A chromatographic analysis was added at the end to determine the final hydrocarbons in lines 287-294 page 9.

Reviewer 2 Report

The work entitled “Biorecovery of agricultural soil impacted by waste motor oil with Phaseolus vulgaris and Xanthobacter autotrophicus”, authors are Blanca Celeste Saucedo Martínez, Liliana Márquez Benavides, Gustavo Santoyo Pizano, and Juan Manuel Sánchez-Yáñez, is devoted to the actual problem of remediation of hydrocarbon contaminated soil. In laboratory experiments, authors used a complex approach to remediate soil contaminated with waste motor oil. This approach included biostimulation with surface-active agents and additional mineral solution, bioaugmentation with hydrocarbon oxidizing Xanthobacter autotrophicus, and phytoremediation with Phaseolus vulgaris. An amazing result was obtained: The level of contamination was successfully reduced from 60,000 ppm to 190 ppm after 3 months.

Major revisions

1) Non-ionic synthetic surfactants, such as Tweens and Triton X-100, are discredited in terms of their toxicity and biodegradability. Their higher toxicity towards microorganisms and EC50 test organisms and incompatibility with the environment in comparison with biosurfactants is mentioned in the literature (for example https://doi.org/10.1016/j.ecoenv.2018.01.019, https://doi.org/10.3390/pharmaceutics13081172). Could authors give some comments about toxicity and biosafety of Triton X-100 and Tween 80? Why did authors not use biosurfactants?

2) It is not entirely clear how phytoremediation is performed. In Materials and Methods, treatment of seeds with X. autotrophicus and further sowing of contaminated soil is described, while in the text, the variant with phytoremediation using P. vulgaris untreated with X. autotrophicus giving the highest level of decontamination (Table 6, T2) is mentioned.

3) How many replicates were used in all experiments? Good statistical proofs are needed to be added into the manuscript, such as means ± standard deviations (errors) and p-values.

4) It is strongly recommended to check, if all Latin names are italicized (it is lost in table captions, for example).

5) It is strongly recommended to check English grammar. Is it correct, for example, to write “stadistically”, not “statistically”? In another example, the letter “c” is lost in the word “distinct” (Table 3).

6) Could authors describe what upper a,b,c,d mean in Tables?

7) What waste motor oil was used? What X. autotrophicus strain was used (isolated by authors, or obtained from any official microbial collection)? What was purity of Tween 80 and Triton X-100? Where the surfactants used were manufactured? This information is necessary to be added in Materials and Methods.

Summary

Reconsider after major revision.

Author Response

Please also see the attachment

Point 1. Non-ionic synthetic surfactants, such as Tweens and Triton X-100, are discredited in terms of their toxicity and biodegradability. Their higher toxicity towards microorganisms and EC50 test organisms and incompatibility with the environment in comparison with biosurfactants is mentioned in the literature (for example https://doi.org/10.1016/j.ecoenv.2018.01.019, https://doi.org/10.3390/pharmaceutics13081172). Could authors give some comments about toxicity and biosafety of Triton X-100 and Tween 80? Why did authors not use biosurfactants?

Response 1. Information on the use of the detergents used and their toxicity are mentioned on lines 236-237 page 8. These detergents have been widely used in the literature and it has been proven that at low concentrations they do not cause any toxic effect to the environment and that they also improve the solubility of hydrocarbons to facilitate their mineralization by microorganisms. At this point of the research, biodetergents have been synthesized but their characterization is still pending.

Point 2. It is not entirely clear how phytoremediation is performed. In Materials and Methods, treatment of seeds with X. autotrophicus and further sowing of contaminated soil is described, while in the text, the variant with phytoremediation using P. vulgaris untreated with X. autotrophicus giving the highest level of decontamination (Table 6, T2) is mentioned.

Response 2. The phytoremediation process is described in detail on lines 268-282, page 9.

Point 3. How many replicates were used in all experiments? Good statistical proofs are needed to be added into the manuscript, such as means ± standard deviations (errors) and p-values.

Response 3. The number of replicates used was clarified, and details of the statistical tests used were added at the bottom of the tables and in the tables.

Point 4. It is strongly recommended to check, if all Latin names are italicized (it is lost in table captions, for example).

Response 4. It was corroborated that the scientific names were written in italics throughout the text.

Point 5.  It is strongly recommended to check English grammar. Is it correct, for example, to write “stadistically”, not “statistically”? In another example, the letter “c” is lost in the word “distinct” (Table 3).

Response 5. English grammar was checked throughout the text.

Point 6. Could authors describe what upper a,b,c,d mean in Tables?

Response 6. The meaning of the letters abc in the tables was explained.

Point 7. What waste motor oil was used? What X. autotrophicus strain was used (isolated by authors, or obtained from any official microbial collection)? What was purity of Tween 80 and Triton X-100? Where the surfactants used were manufactured? This information is necessary to be added in Materials and Methods.

Response 7.  Details of the oil used have been added on lines 232, 233, page 8. Added details of the bacteria on lines 245-248, page 8. Detergent information was added on lines 234-236 page 8.

Round 2

Reviewer 1 Report

The authors made corrections as suggested by the reviewer

Reviewer 2 Report

Please, pay attention on lines 226-227: a part of the sentence is lost.
